# Association between Exposure to Ambient Air Pollution and Age-Related Cataract: A Nationwide Population-Based Retrospective Cohort Study

**DOI:** 10.3390/ijerph17249231

**Published:** 2020-12-10

**Authors:** Jinyoung Shin, Hyungwoo Lee, Hyeongsu Kim

**Affiliations:** 1Department of Family Medicine, Konkuk University Medical Center, Seoul 05030, Korea; jyshin@kuh.ac.kr; 2Department of Ophthalmology, Konkuk University Medical Center, Seoul 05030, Korea; hwlee@kuh.ac.kr; 3Department of Preventive Medicine, Konkuk University School of Medicine, Seoul 05030, Korea

**Keywords:** air pollution, cataract, particulate matter, insurance claim review

## Abstract

This study aimed to investigate the association between ambient air pollutants and cataracts in the general population aged 50 years or older using data from the Korean National Insurance Service—National Sample Cohort. Cataract patients were defined as those diagnosed by a physician and having undergone cataract surgery. After matching the average concentrations of PM_2.5_, PM_10_, NO_2_, CO, SO_2_, and O_3_ in residential areas, the association between quartile level of air pollutants and incidence of cataract was analyzed using a multivariate Cox-proportional hazard risk model. Among the 115,728 participants, 16,814 (14.5%) were newly diagnosed with cataract and underwent related surgery between 1 January 2004, and 31 December 2015. Exposure to PM_10_, NO_2_, and SO_2_ was positively associated with cataract incidence, while O_3_ was negatively associated. The adjusted hazard ratio (HR) with 95% confidence interval was 1.069 (1.025–1.115) in PM_10_ and 1.080 (1.030–1.133) in NO_2_. However, the association between cataract and the quartile of PM_2.5_ measured during one year in 2015 was not clear. The HR of female participants aged 65 or older was significantly increased according to quartile of air pollutants. We identified exposure to PM_10_, NO_2_, SO_2_, and O_3_ associated with cataract development in Korean adults aged ≥ 50 years. This information may be helpful for policymaking to control air pollution as a risk factor for eye health.

## 1. Introduction

Cataract, opacification of the ocular lens, is one of the major causes of loss of useful vision [1]. Globally, the number of people with a severe visual impairment from cataract is projected to be 220 million in 2020 [2]. Blindness due to cataract increased from 10.9 million to 12.6 million between 1990 and 2015 [3]. The prevalence of blindness due to cataracts was found to vary by geographic region, ranging from 12.7% in North America to 42.0% in Southeast Asia [4]. In an analysis using insurance claims data, senile cataract was the most prevalent cause of hospital admission in South Korea in 2019 (*n* = 345,853) [5]. As a leading public health issue, a cataract will become more important as the population increases and life expectancy is extended worldwide [1].

Ambient air pollution, such as particulate matter (PM) < 10 μm in size (PM_10_) and < 2.5 μm (PM_2.5_), nitrogen dioxide (NO_2_), carbon monoxide (CO), sulfur dioxide (SO_2_), and ozone (O_3_), has been recognized as one of the most serious health issues worldwide. Exposure to air pollutants may impact eye health, affecting allergic conjunctivitis [6], dry eye disease [7,8], and age-related macular degeneration [9]. However, these associations are heterogeneous. For example, PM_2.5_ increased allergic conjunctivitis from May to July in Japan, while NO_2_, CO, and oxidants (O_x_) were not associated with that condition [6]. Dry eye disease in Taiwan was associated with increasing levels of NO_2_ and CO over exposure for 10 years [7]. In contrast, in Korean adults, the SO_2_ level was associated with dry eye disease, while exposure to NO_2_, CO, O_3_, and PM_10_ was not [8].

The effect of long-term exposure to air pollution on the development of cataracts is not evident. In India, women aged ≥ 60 years have an increased risk of cataract due to exposure to biomass fuels [10]. Cumulative cadmium exposure increased cataract risk in U.S. adults aged ≥ 50 years, from 1999 to 2008 [11]. Among 18,622 Korean adults > 40 years of age, evidence for an association between air pollutants and cataract was found in O_3_ as a protective effect; the prevalence of cataract was not associated with the levels of PM_10_, NO_2_, and SO_2_ between 2010 and 2012 [12]. In our review of the relevant literature, we found that the results may be affected by differences in race, exposure duration, concentration of pollutants, lifestyle, and known risk factors of cataract, including age, genetic influence, exposure to ultraviolet light, smoking, and diabetes [13,14,15]. Therefore, evaluating the association between exposure to ambient PM_2.5_, PM_10_, NO_2_, CO, SO_2_, and O_3_ and age-related cataract is meaningful for advanced public health care using a representative national database. 

## 2. Materials and Methods 

### 2.1. Data Sources

Data from the Korean Health Insurance Service—National Sample Cohort (NHIS—NSC) were used in this study. The majority of the Korean population (97.2%, approximately 50 million individuals) are covered by the mandatory social National Health Insurance Service. Among them, we selected a random sample of 1 million people (representing about 2% of the total population in 2002, *n* = 1,025,340) after stratification according to age, sex, income level, and type of health insurance (national health insurance and medical aid program). NHIS provides free biennial health examinations to its members aged ≥ 40 years. Approximately 72% of all eligible persons undergo these examinations [16]. In this study, we also used the results of these health examinations to obtain smoking variables. 

The sample cohort resided in 16 regions in Korea, and the change of initial residence was 0–0.3% during the follow-up years [17]. The difference in average health insurance level, which was estimated by income level, was negligible during the cohort years [17]. The NHIS claims data included information on diagnoses, procedures, and prescriptions for inpatient and outpatient visits as identified by the International Classification of Diseases, Tenth Revision (ICD-10) codes and the Korean Drug and Anatomical Therapeutic Chemical Codes.

### 2.2. Study Participants

This study investigated the association between senile cataract in persons aged ≥ 50 years and their exposure to air pollutants. Exclusion criteria were (1) <50 years old in 2002 (*n* = 765,221), (2) death during the research period (*n* = 48,022), (3) change of residence (*n* = 37,237), (4) missing data on income level (*n* = 5465), and (5) unmatched with residence area (*n* = 36,714). Among the remaining 132,681 persons, 13,275 with missing data on smoking status and 3678, who had a cataract diagnosis in 2002–2003, were excluded. Ultimately, 115,728 people participated in this study (Figure 1). A newly developed cataract was defined as a cataract diagnosis (H25, H26) and cataract surgery (S5119) from 1 January 2004, to 31 December 2015, after excluding the patients with a diagnosis of cataract from 2002 to 2003. To define the subject more concisely, we used the definition of cataract, considering both the diagnosis code and treatment code. The index date was defined as the date of the first diagnosis of cataract. 

This study was approved by the Institutional Review Boards (IRBs) of the Clinical Research Ethics Committee of Konkuk University Medical Center, Seoul, Korea (KUH 2019-05-017), and informed consent requirements were waived due to the use of only de-identified data.

### 2.3. Air Pollutant Variables

We obtained the average concentrations of PM_10_, NO_2_, CO, SO_2_, and O_3_ measured hourly from the Korean Air Pollutants Emission Service in 2002–2015. Because PM_2.5_ has been measured in Korea since 2015, we applied the average concentration of PM_2.5_ measured hourly during one year. The air pollutants were measured at 268 nationwide surveillance stations covering most residential areas. Residential five-digit codes that classified “Si (city)” by the first two digits and “Gun,” or “Gu” (town) by the following three digits were used to match the location of the air pollution surveillance stations. We calculated the quartiles of the average yearly concentrations of all pollutants.

PM_10_ and PM_2.5_ were measured using a β-ray attenuation system (PM-711D, Dongil Greensys, Seoul, Korea). NO_2_ was measured using a chemiluminescence instrument (CM2041, APM Engineering Co., Ltd., Gyeonggi-do, Korea). CO was measured using a non-dispersive infrared sensor (ZKJ, Dongil Greensys, Seoul, Korea). SO_2_ was measured using an ultraviolet (UV) fluorescence system (CM2050, APM Engineering Co., Ltd., Gyeonggi-do, Korea). Measurements of all air pollutants were performed according to the standard operating procedures of the Korean Air Pollutants Emission Service of the National Institute of Environmental Research (Incheon, South Korea). The levels of air pollutants and data of meteorological parameters, including annual average temperature, total rainfall, and wind speed, are presented in Appendix A

### 2.4. Other Variables

Participant age was calculated from the birth year until 2002. Health insurance status was divided into 11 categories, 10 national health insurance plans and medical aid based on income status. It was further divided into two groups: lower levels 1–5 in national health insurance and the medical aid level; upper levels 6–10. The residential areas were classified into two groups: Seoul and the six largest cities as “highly urbanized,” and other areas as “less urbanized.” Information on comorbidity was obtained by physician diagnosis using ICD-10 codes before the first diagnosis of cataract: diabetes mellitus (E10–E14); cerebrovascular disease (I63, I64); peripheral vascular disease (I73); chronic pulmonary disease (J44); congestive heart failure (I50); myocardial infarction (I21, I22); malignancy including solitary organ, leukemia, and lymphoma (C00–C97); hemiplegia (G81–G83); liver disease (K74); and chronic kidney disease (N18) [18]. Mental disorders were classified as sensitive information and were masked (F*), and because of this, they could not be differentiated in detail. 

### 2.5. Statistical Analyses

Continuous variables are presented as mean with standard deviation, and the categorical variables are presented as number and percentage of participants. We compared the characteristics of study subjects according to the diagnosis and surgery of cataract using the *t*-test and Chi-square test. The associations between a per-interquartile increase of air pollutants PM_2.5_, PM_10_, NO_2_, SO_2_, CO, and O_3_ from 2002 to 2015 and the incidence of cataract were evaluated using the Cox-proportional hazard regression model after adjusting for age, sex, smoking status, insurance level, urbanization, and comorbidity. We conducted stratified analyses by sex because several previous studies have suggested sex difference in cataract development [4,19]

Additionally, to clarify the effect of age on cataract development, we performed stratified analyses divided by age (younger or older than 65 years). We determined the annual trend in levels of PM_10_, NO_2_, SO_2_, CO, and O_3_ and the number of diagnosed cataracts according to the quartile of PM_10_ from 2002 to 2015. All statistical analyses were conducted using SAS software 9.4 (SAS Institute Inc., Cary, NC, USA). *p* < 0.05 was considered to indicate statistical significance.

## 3. Results

### 3.1. Demographic Characteristics 

As shown in Table 1, 16,814 patients (14.5% of the total subjects) with cataract were newly diagnosed and treated. The breakdown, according to sex, was 37.4% men and 62.6% women. The cataract patient group was older, more likely to have higher-level insurance, and more likely to live in an area other than Seoul and the six largest cities. Diabetes, cardiovascular disease, mental disorder, and chronic kidney disease were identified in a larger proportion in the cataract group than in the no-cataract group. There was no difference in the comorbid rate of malignancy, hemiplegia, or liver disease. 

Data were presented as mean± standard deviation or number (percentage). *p* values were obtained by *t*-test or chi-square test between the cataract group and no cataract group. The lower insurance group was composed of subjects with a low income below the median and medical aid. The upper insurance group was composed of subjects with a high income above the median.

### 3.2. Association of Newly Developed Cataract and Air Pollutants

We found a positive association between PM_10_, NO_2_, SO_2_, and the hazard ratio (HR) of cataract after adjusting for age, sex, smoking status, income level, urbanization, and comorbidity (Table 2). However, O_3_ was negatively associated with the HR of cataract, and PM_2.5_ and CO were not associated with cataract in the total population. Increased quartiles of NO_2_ increased the cataract incidence rate, while increased quartiles of O_3_ decreased it. However, the effect of PM_10_ and SO_2_ on the development of cataract was not dose-dependent.

In a subgroup analysis according to sex, we identified a distinctive association between female patients and cataract incidence. The association in males was weak with PM_10_ and non-existent with NO_2_, SO_2_, CO, and O_3_; in females, these associations remained consistent. In a subgroup analysis, according to age (65 years old), the associations between PM_10_, NO_2_, SO_2_, and O_3_ and incidence of cataract were significant only in subjects ≥ 65 years. 

### 3.3. The Annual Trends of Mean Concentration of Air Pollutants and Number of Diagnosed Patients

The upper graph in Figure 2 shows the annual mean concentrations of PM_10_, NO_2_, SO_2_, CO, and O_3_ from 2004 to 2015. The annual mean concentrations of PM_10_ gradually decreased in the later period, but those of NO_2_, SO_2_, CO, and O_3_ increased or did not change. The lower bar graph in Figure 2 shows the number of diagnosed patients according to the quartile of PM_10_. According to decreased annual mean concentrations of PM_10_, the total number of diagnosed patients decreased from 2004 to 2015. However, there was no proportional trend in cataract incidence according to the quartile of PM_10_.

## 4. Discussion

In this nationwide, retrospective, population-based Korean cohort study of claims data, 14-year exposure to PM_10_, NO_2_, and SO_2_ was positively associated with cataract incidence, and exposure to O_3_ was negatively associated. The effect of air pollutants was more evident in females and subjects aged ≥ 65 years. This study showed that long-term exposure to PM_10_, NO_2_, and SO_2_ might be a risk factor for the development of cataract in the general population aged ≥ 50 years.

A cataract develops from various causes: metabolic disorder, nutritional deficiency, or environmental stressors, such as extreme low or high temperature, radiation (UVB, X-ray, infrared, sunlight), metal ions (mercury, copper, lead), and toxins (acetaldehyde, bisphenol A, smoking) [20]. Although this study could not determine the mechanism, long-term exposure to air pollutants as a cataractogenic stressor may damage the membrane luminal and secretory proteins by oxidative stress from reactive oxygen and nitrogen species [12,20]. Lens opacity from cumulative damage of environmental insults could be accelerated by aging [21]. Additionally, the modification of the lens and weakness of the protective mechanisms against stress that occurs with increasing age may contribute to the development of cataract [21,22]. Because we found that older subjects had a high risk of cataracts from exposure to air pollution in this study, a basis has been laid for reducing exposure to air pollutants as an oxidative stressor in older people. 

It is argued that air pollution absorbs UVB photons and reduces the amount of solar UVB radiation reaching the Earth’s surface [23]. However, because PM density, composition, and shape vary spatiotemporally, it is difficult to determine their effects on UV radiation [24]. In general, PM comes from secondary aerosol sources via an atmospheric chemical reaction of gaseous pollutants, such as NO_2_ and SO_2_, resulting from fuel combustion. PM also includes various suspended solid particles and liquid droplets that originate from organic or inorganic sources. The hazardous level of PM was determined by the proportions of organic components, such as soil dust or a natural source, and inorganic components, such as motor vehicle operation, biomass/field burning, combustion/industry, or secondary aerosol [25]. Based on the variance of the components, it is thought that cataract incidence is more affected by PM_10_ than by other pollutants. The association between PM_2.5_ and cataract incidence could not be confirmed in this study, because the difference in PM_2.5_ concentration over one year was insufficient to reflect the effects on cataract incidence. The PM_2.5_ concentration measured over one year showed no significant difference in regional distribution (mean: 23.1 μg/m^3^; min: 16.0 μg/m^3^; max: 30.0 μg/m^3^). Therefore, further research is needed to compare the effects of long-term exposure of PM_2.5_ and PM_10_ on cataract incidence in a large population-based cohort after controlling the confounding factors.

Differences in rates of cataract by sex were shown in previous studies [14,19]. In the Global Burden of Disease Study 2015, age-standardized, disability-adjusted life year (DALY) rates were 54.5 among men vs. 65.0 among females in 1990, and 52.3 among men vs. 67.0 females in 2015. Females had higher rates of cataract than males of the same age [19]. The mixed effect of exposure to indoor cooking fuels, outdoor activity, and hormone replacement therapy on cataract should be considered in higher-risk females [10,26,27]. This study adds to existing research on the significant effect of air pollutants on females.

Exposure to ozone is negatively associated with the HR of cataract, similar to another Korean study [12]. In general, exposure to ozone results in the deterioration of the ocular surface and an inflammatory state [28]. However, ozone characteristically does not easily penetrate the cornea due to polarity. Therefore, ozone-related oxidative damage may not directly affect the lens [12]. Although hazards to eye health have been identified in short-term exposure of O_3_, the risks may not increase for long-term exposure of O_3_, even at similar concentrations [29]. This may be because the O_3_ level in the Kim et al. study (below 40 ppm) [29], which was measured in Incheon, South Korea, was similar to the mean level of O_3_ in our data (max 40 ppm), which did not reach the levels of National Ambient Air Quality Standards (8 hr exposure to 70 ppm) or WHO air quality guidelines (daily maximum 8-hr mean 100 μg/m^3^) [30]. UV irradiance does not follow the ozone trend, and an increased level of tropospheric ozone may protect the lens from UV exposure [23]. Ozone showed a reverse association with PM_10_, NO_2_, SO_2_, and humidity in dry eye disease [31]. Although we can explain the protective mechanism of ozone on the incidence of cataract in this study, clinical and experimental studies on the relationship are needed to consider a direct role of O_3_ or a mediator of PM_10_, NO_2_, and SO_2_ with a reverse association.

A linear relationship to the quartiles of pollutants and cataract incidence was not found in this study. We believe that the reason for this finding is a threshold effect, which is that the hazardous effect of air pollutants may not be obvious at low levels of air pollutants below any cut-off value [15]. The levels of National Ambient Air Quality Standards for PM_10_ and PM_2.5_ are 150 μg/m^3^ and 35 μg/m^3^, respectively [30], which is higher than the maximum values measured in this study (PM_10_: 61.0 μg/m^3^ and PM_2.5_: 30 μg/m^3^). Therefore, the linear relationship to the quartiles of pollutants and cataract incidence could not be clearly seen below a cut-off concentration of air pollutants, which was not high enough to have an obvious effect. 

Characteristics of subjects according to residential area or insurance status show a mixed pattern in cataract incidence. Due to a large number of subjects, there may have been significant statistical differences according to residential area or insurance status, although the difference is clinically meaningless. While cataracts can be predicted to be low due to a reduction of UV irradiance in highly polluted urban areas, eye health risks can be high due to the influence of job or smart device utilization rates in urban areas [23]. Because most of Korea’s territory is mountainous, the distribution of the population is uneven and the degree of urbanization varies, indicating the difference in air pollution concentration. Furthermore, the air pollution surveillance station and ophthalmic clinic for diagnosis and treatment may be arranged along with urban areas or large cities. Therefore, it is necessary to consider the levels of urbanization when you interpreted these results.

A higher risk of cataract was reported in people with low socioeconomic status [14]. However, the proportion of high-insurance-level subjects in the cataract group was high in this study. We think this is because we considered both diagnosis and surgery of cataract in the target population, which may be affected by socioeconomic status.

Though smoking is a known risk factor for cataracts [2], cataract incidence was higher in non-smokers than in smokers in this study. Although the study did not show the smoking rate among women, the smoking experience rate for Korean women aged > 40 is reported to be 2.5% [32]. Because the smoking rate in women was low while the cataract incidence in women was higher than that in men, it could be misinterpreted as if smoking has the protective effect of cataract incidence. Therefore, care needs to be taken when interpreting this result. 

This study had several limitations. First, we could not obtain any information on the grade or type (nuclear, cortical, or posterior sub-capsular) of cataract in this database and so did not compare these associations according to grade or type of cataract and air pollution, although the risk factors among types of age-related cataract may be different [22,23]. However, because we defined the study subjects as those who underwent cataract surgery, we can substitute the severity of cataract for select subjects. The study identified the incidence of age-related cataract over 50 years of age, but the age limit should consider the possibility of selection bias. Over the years, the study population got older and maybe reached an age where the cataract is diagnosed less often. Although several previous studies have selected people age 50 or older [13,27,33], we think a consensus of age limitation is necessary. Second, we matched the residence and local air pollutant levels. Therefore, if participants worked at a distant location from their domiciles, our matching system would not have accurately reflected the exposed air pollutant levels. There is a possibility of information bias because subjects were excluded if their residence was changed or not matched. Furthermore, cataract patients diagnosed early in the study did not reflect exposed air pollutants because the average concentration of air pollutants was calculated from 2002 to 2015. Although the long-term exposure of air pollutants is expected to impact significantly, the levels of air pollutants before 2002 cannot be obtained. Therefore, there is a limit to linking prolonged exposure before diagnosis in the subjects who were diagnosed early period. Third, we could not obtain data on traffic-related air pollution or indoor air pollution, which may be the main source of PM in Korea. Additionally, a multiple-pollutant model has been applied because the associations among air pollutants, such as a co-linearity, would have affected it. Last, due to the lack of information about economic status, outdoor activity related to sun exposure, labor effect, and ophthalmic clinic accessibility, we could not consider those possible effects on the results [4,33]

Despite these limitations, the strength of our study is that the effects of long-term exposure of PM_10_, NO_2_, SO_2_, CO, and O_3_ were reported in the newly developed cataract in the representative subjects. Further study is needed to identify the health effects of other important environmental pollutants that are not well-known, such as asbestos [34,35] and the impact of exposure to air pollutants over the decades, without age restrictions on the subjects.

## 5. Conclusions

Long-term exposure to PM_10_, NO_2_, SO_2_, and O_3_ was associated with cataract development in Korean adults aged ≥ 50 years. Especially, cataract incidence has been affected by the annual mean concentrations of PM_10_. The association between air pollutants and cataract incidence differed according to age (65 years old) and sex. This information may be helpful for understanding eye health and policymaking to control air pollution.

## Figures and Tables

**Figure 1 ijerph-17-09231-f001:**
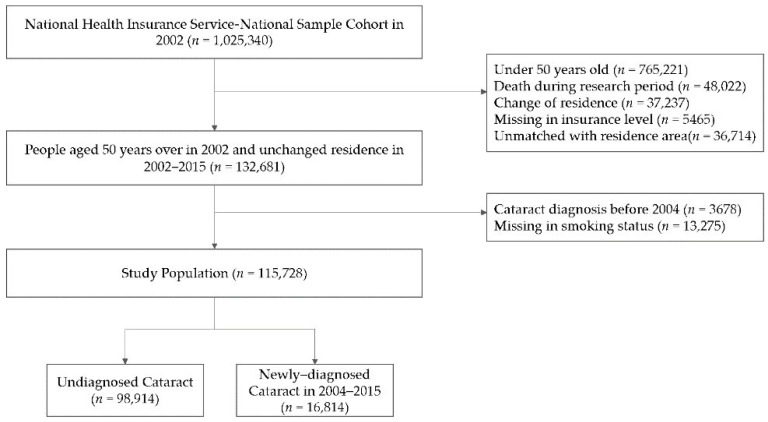
Flow chart of study population.

**Figure 2 ijerph-17-09231-f002:**
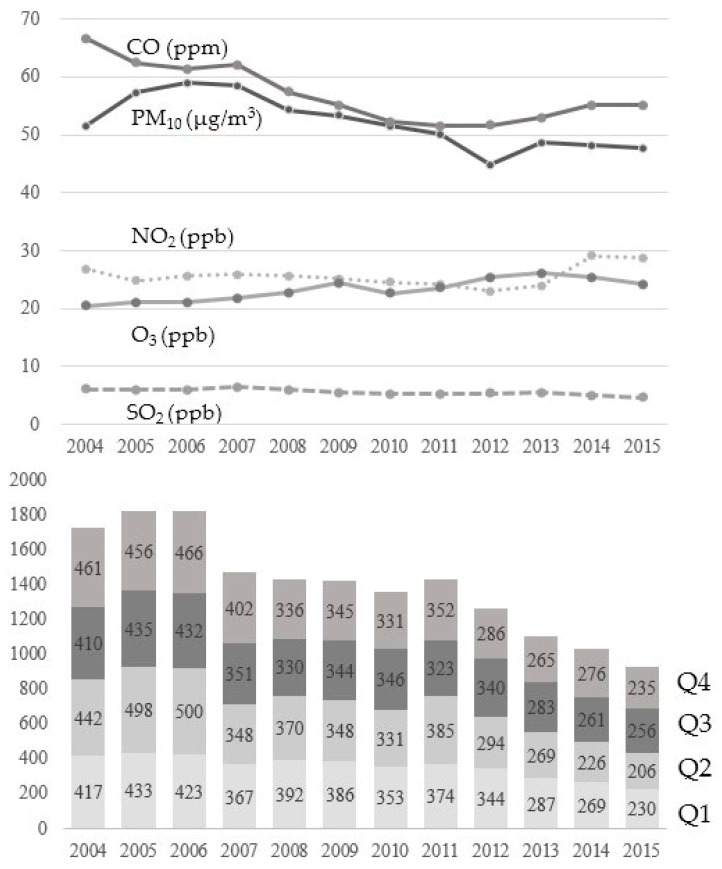
Trend in the levels of air pollution (**upper**) and the patients’ number of diagnosed cataract according to the quartile levels of PM_10_ from 2004 to 2015 (**lower**).

**Table 1 ijerph-17-09231-t001:** Baseline characteristics of study population.

	Total (*n* = 115,728)	No Cataract (*n* = 98,914)	Cataract (*n* = 16,814)	*p*-Value
Age, year	60.0 ± 7.2	59.4 ± 7.5	63.2 ± 6.9	<0.001
Sex				<0.001
Male	54,679 (47.2)	48,395 (88.5)	6284 (11.5)	
Female	61,049 (52.8)	50,519 (82.8)	10,530 (17.2)	
Smoking status				<0.001
Non-smoker	86,453 (74.7)	72,860 (84.3)	13,593 (15.7)	
Former smoker	9663 (8.4)	8429 (87.2)	1234 (12.8)	
Current smoker	19,612 (16.9)	17,625 (89.9)	1987 (10.1)	
Insurance levels				<0.001
Lower	45,884 (39.6)	39,319 (85.7)	6565 (14.3)	
Upper	69,844 (60.4)	59,595 (85.3)	10,249 (14.7)	
Urbanization				<0.001
High	50,207 (43.4)	43,382 (86.4)	6825 (13.6)	
Less	65,521 (56.6)	55,532 (84.8)	9989 (15.2)	
Comorbidity				
Diabetes mellitus	22,314 (19.3)	17,734 (17.9)	4580 (27.2)	<0.001
Cerebrovascular disease	4360 (3.8)	4200 (4.2)	1060 (6.3)	<0.001
Peripheral vascular disease	4203 (3.6)	3347 (3.4)	856 (5.1)	<0.001
Chronic pulmonary disease	2712 (2.3)	2172 (2.2)	540 (3.2)	<0.001
Mental disorder	2243 (1.9)	1776 (1.8)	467 (2.8)	<0.001
Congestive heart failure	1449 (1.3)	1153 (1.2)	296 (1.8)	<0.001
Myocardial infarction	599 (0.52)	500 (0.51)	99 (0.59)	<0.001
Malignancy	550 (0.48)	475 (0.48)	75 (0.45)	0.516
Hemiplegia	533 (0.46)	454 (0.46)	79 (0.47)	0.848
Liver disease	409 (0.36)	330 (0.33)	79 (0.47)	0.526
Chronic kidney disease	314 (0.27)	254 (0.26)	60 (0.36)	0.021

**Table 2 ijerph-17-09231-t002:** Hazard ratios (HR) of cataract incidence according to the levels of air pollutants.

		Total (*n* = 115,728)				Male (*n* = 54,679)				Female (*n* = 61,049)				<65 Years Old (*n* = 101,372)				≥65 Years Old (*n* = 14,356)			
		Adjusted HR	95% CI		*p* Value for Trend	Adjusted HR	95% CI		*p* Value for Trend	Adjusted HR	95% CI		*p* Value for Trend	Adjusted HR	95% CI		*p* Value for Trend	Adjusted HR	95% CI		*p* Value for Trend
PM_10_	Q1	1			<0.001	1			0.028	1			<0.001	1			0.106	1			<0.001
	Q2	1.071	1.026	1.118		1.076	1.004	1.154		1.068	1.012	1.127		1.011	0.954	1.070		1.129	1.059	1.203	
	Q3	0.992	0.951	1.036		1.016	0.947	1.089		0.980	0.928	1.035		0.946	0.894	1.010		1.031	0.966	1.101	
	Q4	1.069	1.025	1.115		1.094	1.021	1.173		1.058	1.004	1.116		0.998	0.943	1.055		1.143	1.071	1.218	
PM_2.5_	Q1	1			0.547	1			0.891	1			0.646	1			0.475	1			0.437
	Q2	1.012	0.833	1.230		0.934	0.679	1.284		1.039	0.810	1.314		0.907	0.720	1.142		1.387	0.939	2.049	
	Q3	0.927	0.760	1.130		1.001	0.752	1.333		0.866	0.657	1.141		0.883	0.706	1.105		1.127	0.719	1.766	
	Q4	0.905	0.772	1.062		0.917	0.717	1.784		0.903	0.731	1.115		0.873	0.724	1.054		1.079	0.791	1.471	
NO_2_	Q1	1			0.011	1			0.122	1			0.001	1			0.699	1			0.004
	Q2	1.035	0.993	1.079		1.053	0.984	1.128		1.026	0.974	1.085		1.025	0.969	1.084		1.043	0.981	1.108	
	Q3	1.062	1.015	1.112		1.070	0.993	1.153		1.059	1.000	1.122		1.029	0.969	1.093		1.102	1.027	1.181	
	Q4	1.080	1.030	1.133		1.097	1.014	1.186		1.072	1.009	1.139		1.036	0.973	1.103		1.137	1.056	1.224	
SO_2_	Q1	1			0.026	1			0.732	1			0.031	1			0.332	1			0.025
	Q2	1.065	1.021	1.111		1.010	0.942	1.083		1.102	1.044	1.162		1.044	0.986	1.106		1.086	1.019	1.157	
	Q3	1.046	1.002	1.091		1.033	0.964	1.108		1.055	0.999	1.113		1.004	0.949	1.063		1.090	1.022	1.163	
	Q4	1.027	0.984	1.073		1.035	0.963	1.112		1.026	0.971	1.083		0.993	0.937	1.052		1.067	0.998	1.141	
CO	Q1	1			0.064	1			0.286	1			0.033	1			0.409	1			0.091
	Q2	1.006	0.964	1.049		0.946	0.882	1.014		1.043	0.989	1.100		1.000	0.946	1.058		1.013	0.949	1.080	
	Q3	1.047	1.004	1.093		1.007	0.939	1.079		1.072	1.015	1.132		1.043	0.986	1.104		1.057	0.989	1.129	
	Q4	0.991	0.949	1.035		0.979	0.911	1.051		1.001	0.948	1.057		1.016	0.959	1.076		0.969	0.907	1.036	
O_3_	Q1	1			0.013	1			0.199	1			0.022	1			0.226	1			0.017
	Q2	0.991	0.947	1.038		0.997	0.927	1.074		0.990	0.933	1.049		0.993	0.935	1.053		0.989	0.921	1.063	
	Q3	0.997	0.953	1.044		1.000	0.928	1.077		0.998	0.941	1.058		1.007	0.948	1.069		0.984	0.916	1.057	
	Q4	0.931	0.888	0.977		0.936	0.866	1.012		0.928	0.874	0.986		0.952	0.893	1.014		0.908	0.844	0.977	

HRs and 95% confidence interval (CI) were obtained by Cox-proportional hazard regression analysis after adjusting for age, sex, smoking status, income levels, urbanization, and comorbidity.

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
