# Peer review of "Association between Exposure to Ambient Air Pollution and Age-Related Cataract: A Nationwide Population-Based Retrospective Cohort Study"

_ijerph, 2020, doi:10.3390/ijerph17249231_

Round 1

Reviewer 1 Report

It is a well thought out and well-presented paper. Just two small observations:

1) Did the Authors consider the labor effect as an independent exposure factor? I believe that at least this aspect should be investigated in terms of a possible confounding factor on the results presented.

2) Authors present a comparison between urbanized and less urbanized areas, but this data is really little valued in terms of comparison, considering how much it can actually weigh in terms of variation in the level of pollutants. Is there a particular reason why this variable is only inserted into the model?

Author Response

  We are pleased to give us a chance to revise our manuscript and further consideration. We did our best in revising the manuscript following comments raised by the reviewers. We look forward to hearing further. We appreciate the valuable comments.

It is a well thought out and well-presented paper. Just two small observations:

1) Did the Authors consider the labor effect as an independent exposure factor? I believe that at least this aspect should be investigated in terms of a possible confounding factor on the results presented.

Answer) I’m not sure if we understand “labor effect” correctly, but if it means cataract or air pollution from labor, we didn’t consider it. This study did not measure the labor effect by analyzing claims data. Therefore, this is specified as one of the limitations in the discussion (line 277-278).

2) Authors present a comparison between urbanized and less urbanized areas, but this data is really little valued in terms of comparison, considering how much it can actually weigh in terms of variation in the level of pollutants. Is there a particular reason why this variable is only inserted into the model?

Answer) We agreed that the importance of the comparison between urbanized and less urbanized areas when interpreting these results. Air pollution is high in large cities, but air pollution levels in the western region are generally high in South Korea due to the influence of China and airflow. Local factors have to be considered, such as air pollution surveillance stations and ophthalmic clinics for diagnosis and treatment.

 (Please show pictures in the attached file)

Air pollution surveillance station         Density of Korean population

About 80% of the land in South Korea is mountainous. Due to the characteristics of the terrain, the distribution of the population is uneven and the degree of urbanization varies, indicating the difference in air pollution concentration.

We considered the comparison between urbanized and less urbanized areas in this study because 268 air pollution surveillance stations are located in densely populated areas.

Therefore, this was added to the discussion (line 242-246).

Reviewer 2 Report

In line 24, you mention no effect detected in 2015, this is a little confusing, can you clarify.

In line 47 you refer to 3 years of exposure, surely there is a life-time of exposure (I know this is from a cited paper).

I find table 1 detailed but a little confusing, what the p-values refer to, eg if we go to smoking as an example;

73% of those with No cataract are non-smokers, but of those with a cataract 80% are non-smokers, suggesting smoking is good!

Also if I go further down the table and say look at mental disorder, many with mental disorder have a smoking habit, have these been controlled for within the analysis, its not clear.  So I suppose what I`m trying to ask related to co-morbities and how these are included or excluded within he analysis.

It appears that the percentage values in some places, mean something different in other places within the table.

I`d like more explanation as to the thinking as to why there is no observed effect for PM2.5 and there is for PM10, this needs to be considered in more detail.

PM10 and NO2 show a lot of colinearlity, has this been controlled for within the models? were univariate and multi-variate analysis conduced to see how it affected the estimates for a given pollutant.

Author Response

We are pleased to give us a chance to revise our manuscript and further consideration. We did our best in revising the manuscript following comments raised by the reviewers. We look forward to hearing further. We appreciate the valuable comments.

  1. In line 24, you mention no effect detected in 2015, this is a little confusing, can you clarify.

Answer) We revised the sentence like that: the association between cataract and the quartile of PM2.5 measured during one-year in 2015 was not clear in line 24. We intended to express that the effect of PM2.5 was not clear because it was measured for one year, unlike other pollutants.

  1. In line 47 you refer to 3 years of exposure, surely there is a life-time of exposure (I know this is from a cited paper).

Answer) The cited study said that environmental factors were obtained from 2010 to 2012 (for three years) and encountered annual average values. We can’t find any evidence to declare the duration of exposure (life-time of exposure). Therefore, we decided to delete the duration (3 years) in line 48. 

  1. I find table 1 detailed but a little confusing, what the p-values refer to, eg if we go to smoking as an example; 73% of those with No cataract are non-smokers, but of those with a cataract 80% are non-smokers, suggesting smoking is good!

Also if I go further down the table and say look at mental disorder, many with mental disorder have a smoking habit, have these been controlled for within the analysis, its not clear. So I suppose what I`m trying to ask related to co-morbidities and how these are included or excluded within he analysis.

It appears that the percentage values in some places, mean something different in other places within the table.

Answer) We also thought a lot about what you mentioned. There has been little research on cataracts of air pollutants in the past, and we wanted to consider a lot of known risk factors. Smoking is an obvious risk factor for cataracts. Nevertheless, we think that this result is as follows:

The smoking rate for Korean women is low, especially for those over 40 years old, at 2.5% (Kim et al., study). On the other hand, cataract incidence is higher in women (62.6%) than men (37.4%). Therefore, we have already described that careful interpretation was needed. To prevent misinterpretation, we divided paragraphs and corrected expressions, additionally in line 251-256.

Total

(n=115,728)

No Cataract (n=98,914)

Cataract (n=16,814)

P- value

Comorbidity

Diabetes mellitus

   Yes

22,314 (19.3)

17,734 (17.9)

 4,580 (27.2)

<0.001

   No

93,414 (80.7)

81,180 (82.1)

12,234 (72.8)

In the comorbidities in Table 1, the only diagnosed person (corresponding to “yes”) are shown.

The p-value is presented after comparing the cataract and no-cataract groups, which is added as a table caption.

   4. I`d like more explanation as to the thinking as to why there is no observed effect for PM2.5 and there is for PM10, this needs to be considered in more detail.

Answer) The difference in the quartile value of PM2.5 is not significant (mean: 23.1 μg/m3; min: 16.0 μg/m3, max: 30.0 μg/m3, 25th 20.3 μg/m3, 50th 22.0 μg/m3) and does not show regional differences. This may be a characteristic of PM2.5, but the authors thought there was a limit because it was measured only for one year. We added this explanation in line 207-209.

  1. PM10 and NO2 show a lot of colinearity, has this been controlled for within the models? were univariate and multi-variate analysis conducted to see how it affected the estimates for a given pollutant.

Answer) As we mentioned in the 198-200 line, PM was known for coming from secondary aerosol sources, such as NO2 and SO2, which originate from fuel combustion. Therefore, it may have a colinearity between PM10 and NO2. This study used a multivariate analysis. However, we did not control the effect of NO2 in the PM10 model. Although previous studies have controlled other air pollutants considering the covariate, our research design did not have any evidence to prove the colinearity of PM10 and NO2 (only confirm a partial correlation). We found the different aspects of PM10 and NO2 in the results according to sex (Table 2). Figure 2 shows that the trends of PM10 and NO2 showed a different feature since 2012; the initial parameter has some error in the unit. Therefore, we revised the sentences of results (3.3. The annual trends of the mean concentration of air pollutants and the number of diagnosed patients). We are sorry to give confusion.

We added a limitation that failed to consider the possibility of co-linearity for PM10 and NO2 (SO2) in 275-276 lines. However, the PM10 analysis considered a confounding factor (NO2 and SO2) was not conducted because of the reasons described earlier.

Each indicator in Figure 2 was clearly marked and the units and the names of each indicator were inserted.

Reviewer 3 Report

Dear Authors,

I carefully evaluated your paper, founding it overall well written and well presented. The paper focuses on a relevant issue that needs to be properly investigated. The scientific background is well presented and the results are relevant. No concerns on methodology. The discussion is clear and the contribution to the literature is also clearly presented.

My only consideration is regarding the introduction, the Authors should discuss the health effects of other important environmental pollutants such as asbestos (i.e. consider: doi: 10.1186/s13104-019-4675-4 ; doi: 10.1016/j.ccm.2020.08.013 )

Author Response

We are pleased to give us a chance to revise our manuscript and further consideration. We did our best in revising the manuscript following comments raised by the reviewers. We look forward to hearing further. We appreciate the valuable comments.

Dear Authors,

I carefully evaluated your paper, founding it overall well written and well presented. The paper focuses on a relevant issue that needs to be properly investigated. The scientific background is well presented and the results are relevant. No concerns on methodology. The discussion is clear and the contribution to the literature is also clearly presented.

My only consideration is regarding the introduction, the Authors should discuss the health effects of other important environmental pollutants such as asbestos (i.e. consider: doi: 10.1186/s13104-019-4675-4 ; doi: 10.1016/j.ccm.2020.08.013 )

 Answer) We added the sentences that we need to study the health effect of other important environmental pollutants that are not well-known, such as asbestos, and referenced the two articles (reference no. 34 and 35). Considering the flow of the manuscript, we described it at the end of the discussion, not the introduction.

Reviewer 4 Report

The paper examines the association between air pollution and cataract risk in a Cox regression analysis. I must confess that I never considered cataract as a disease affected by air quality. I always thought the lens is a tissue with very little metabolic activity and thus likely not much influenced by external factors besides radiation that directly reaches the lens. But I agree that cataract is a disease of old age and chronic oxidative stress can hasten aging processes. I did not know before that smoking is a risk factor for cataract. The authors pointed out literature that seems to point in that direction and if smoking can cause cataract then to me that makes it sound worthwhile to also investigate air pollution as a possible factor, because of similarities between air pollution and smoking.

I therefore was rather curious about this paper. But I am sorry to tell that I did not fully understand the methodology or even the rationale behind the choice of some methods. It seems the authors started off with a random sample of about 1 million citizens. From this random sample they excluded all people aged over 50 years in 2002. Why did they do this? I understand that cataract is rare in younger people. But persons aged 51 in 2020 would also have reached an age with a higher incidence of cataract until the end of follow up. They excluded persons that died between 2002 and 2015. Why was this done? A Cox proportional hazard model can deal with loss to follow up and death. They excluded persons that changed their residence during the follow up. But also for those persons the average exposure during the observation period can be calculated. But in my understanding air pollution effect on cataract risk will not be short-term but rather long-term in the range of decades rather than years. So the exposure during the observation period is certainly of less importance than the (average) exposure over the decades before. Maybe even childhood exposure is relevant. But we have no information on residency at that time. They excluded persons with cataract diagnosis in 2002 and 2003. But it seems they did not have information on cataract diagnosis prior to 2002. A person could have been diagnosed in 2001 and undergo surgery in 2003. That person would have been included in the study.

I believe the authors calculated average exposures over the 14 years from 2002 till 2015. I understand that we are mostly interested in the exposure difference between regions, not in the absolute concentrations. It is important to ascertain that regions with high air pollution levels in the period at or after 2002 had also high air pollution levels in the previous decades. The authors write: “The concentrations of air pollutants were calculated at least two years before cataracts were diagnosed.” This could simply mean that the averaging period started 2 years before the cataracts were counted. Or it means that they calculated exposure for every person individually. If a person did not experience cataract exposure was calculated as the average exposure until 2015. If a person was diagnosed with cataract in 2006 then exposure was calculated as the average from 2002 until 2 years before diagnosis, i.e. 2004. The latter method would mean that persons who do not experience cataract or who experience cataract late in the observation period will be attributed lower pollution levels because of declining pollution levels at least for NO2 and PM10. This would introduce a serious bias. But if, on the other hand, for all persons exposure was defined as the average until 2015 then exposure levels in 2015 would be treated as if the affected cataract diagnosis in 2004. Why not use exposure in the years 2002 and 2003 as proxy as long-term past exposure? That would ensure no bias because of linear temporal trends and also certain temporality between exposure and effect.

It is totally unclear how exposures were calculated for specific persons. It seems region level exposures were estimated based on data from 268 monitoring stations. But it is not a straightforward task to decide which station is representative for which region. It is not even clear for the reader what is meant by “region”. The phrase “Residential five-digit codes that classified “Si,” “Gun,” and “Gu”” is completely illegible for any outsider. How many people live on average in an area defined by residential five-digit codes? Are these regions always homogenous regarding pollution levels? Can every region of this kind be linked to exactly one monitoring station? In the main paper we only learn that pollution concentrations were divided into quartiles. Only in the annex do we learn that pollutants were measured as µg/m³, ppb and ppm respectively and the levels that were measured. Without the information on the measurement units used, the figure 2 upper panel cannot be interpreted correctly.

Numbers in chemical formulas (e.g. O3, NO2) should be written as subscripts. Throughout the text (not always, but often) and also in figure 2 this was not done.

It is not completely clear how insurance groups were handled. It seems there are either 10 or 11 health insurance groups indicative of income or socioeconomic status: 10 national health insurance plans and one medical aid. But then these 11 groups are merged into a binary variable but it is not clear how the medical aid group was handled.

Table 1 indicates a protective effect of smoking. In the discussion the authors argue that this might be due to confounding. In that case they should present the results of their Cox regression for smoking. If the confounders they included in their model could not reverse the protective effect of smoking either the model is biased or the confounders are not sufficient. Similarly we find higher cataract rates in rural areas and also this is not very well explained in the discussion. We find nearly the same percentages of cataract in the two insurance levels (14.3 versus 14.7%). Nevertheless the table claims that difference is highly significant (p < 0.001). Maybe this is because of the large number of persons under study and even irrelevant differences become statistically significant. Or the p-value is incorrect. Also p-values for the co-morbidities are reported. But I do not find an explanation against what state they are tested. Maybe the reference is “no other disease”?

Results, line 158ff: “Although the highest quartiles (Q4) of PM10 and NO2 in males showed higher adjusted HR than that of females, the association in males was weak with M10 [correct: PM!] and non-existent with NO2, SO2, CO, and O3;”: The adjusted HR of the 4th quartile was not only higher than in females, but also significantly higher than for the first quartile (for PM10 and for NO2). Table 2 reports that the linear trend was not significant though. This is mostly because also the HRs for the 2nd and 3rd quartile were substantially higher than that for the 1st quartile. This does not lend credence to the claim (in the discussion) that women are more susceptible to air pollutants than men.

Figure 2 shows declining cataract incidence over the years. Why is this that? Either cataracts are decreasing generally in Korea, or most of the cased are diagnosed between the age of 50 and 60. In the beginning of the observation period most people were in this age range (older than 50 because of definition, not too old because they had to survive until the end of the observation period). Over the years the cohort grew older and maybe reached an age where cataract is diagnosed less often: all individuals that have a (genetic?) risk of acquiring cataract have already been diagnosed before. This would mean that the original decision only do include persons older than 50 in 2002 was misguided!

I suppose also in Korea some pollutants are positively or negatively correlated with each other. Given the large cohort size a multi-pollutant model would have been informative.

Discussion line 184: “environmental stressors such as cold or high temperature, ionizing radiation (UVB, X-ray, infrared, sunlight)”: “cold or high temperatures” is wrong. Either it is “cold and heat” or “low and high temperature”. But I am not sure that cold or “low temperature” is really a risk factor. I have never heard that people living in the arctic or that often swim in cold waters do get cataract! And: UV-radiation might be termed “ionizing” although most will only call high-frequency UVC or higher “ionizing”. But IR certainly is not ionizing (although it is a risk factor for cataract).

Summary: better explain selection of participants and exposure assessment. Report exposures including units of measurement! Report findings of Cox regression also for the other factors (especially smoking) and also show how these factors affected HR estimates of air pollutants. If possible also perform multipollutant models.

Author Response

Please see the attachment to show the figure. 

Round 2

Reviewer 2 Report

The authors had responded appropriately to the points raised by the reviewers.

I note that in Table one the p-value  for the Insurance groups is significant, but the percentage with cataracts in each insurance group is almost identical.  Does this p-value relate to those with and without a cataract in a specific insurance group?

It would be much better to see a comparison between the groups.

I do not know enough about the timeline for cataracts to develop, there seems to be a lot more of them in the earlier years of the study.

If linked to pollution, then exposure prior to the study period might be important.

With time (figure 2) pollutants drop and so do cataract incidence.  Could the drop in cataracts be due to better health systems, awareness etc.  Its not totally convincing that the pollutants are the sole causal component.

The literature suggests indoor air pollutants might be more important from a cataract perspect, has indoor sources been considered and controlled for.

Reviewer 4 Report

You write: “A linear relationship to the quartiles of pollutants and cataract incidence was not found in this study. We believe that this is a threshold effect at low levels of air pollutants [15]. Therefore, the 231 association could not be clearly seen below a cut-off concentration of air pollutants.” I do not understand on which data you base this statement. Table 2 clearly shows for PM10, NO2, SO2, and nearly also for CO a significant (linear) trend. For all these pollutants Q2 shows a higher HR compared to Q1 and this difference is even significant in the case of PM10 and SO2. So I do not see any clear evidence of a threshold effect!

You write: “The air pollutants were measured at nationwide surveillance stations located in residential areas, covering most populated areas. Residential five-digit codes that classified “Si,” “Gun,” and “Gu” were used to match the location of the air pollution surveillance stations.” I have asked you to explain the meaning of 5-digit code areas and the classification as “Si” etc. I also asked you to explain which types of monitoring stations are used (residential background or hot-spot locations). In your response to me you explain that the monitoring stations are residential background stations so that they are representative for a wider area. You do still not provide this information in the paper. And I am still not sure how you assigned air pollution levels to a 5-digit codes area. There might be 5-digit codes areas with no station, there might be areas with exactly one station, or there might be areas that have 2 or more stations. Only in the case of exactly 1 station it would be easy to assign a pollution level to the whole area (although there might be some spatial variation within the area still), but with no station you would have to average the values from different station and you could do this differently, e.g. by expert assessment: which nearby station is most representative for the area? – or you could calculate a distance-weighted average for example.

Summing up I asked you: “better explain selection of participants and exposure assessment. Report exposures including units of measurement! Report findings of Cox regression also for the other factors (especially smoking) and also show how these factors affected HR estimates of air pollutants. If possible also perform multi-pollutant models.” You have explained the rationale of the selection of participants to me. But in the paper you have only mentioned the weaknesses of that selection in the discussion but have not provided any rationale for those choices. As discussed above the description of the exposure assessment is still incomplete. You still do not report the adjusted HRs for smoking. I also believe that the smoking effect in table 1 is confounded by sex. But this needs to be demonstrated in the adjusted model, if not in a separate table then at least in the results text. And you did not comply with my strong suggestion to perform multiple-pollution models. If the spatial distribution of the pollutants demonstrates a strong positive or negative correlation then the findings for one pollutant (e.g. negative effect of ozone) could be confounded by the influence of other pollutants. So either you report the correlation between pollutants or you report the outcome of 2-pollutant models.

You write: “In a subgroup analysis according to sex, we identified a distinctive association between female patients and cataract incidence. The association in males was weak with PM10 and non-existent with NO2, SO2, CO, and O3;…” I still do not see this observation in table 2! Yes, there is no clear linear trend in males. Maybe due to the smaller number of male patients the study power is reduced. But the effect estimates (HR) per quartile are partly even higher in males than in females! E.g. NO2, Q2: 1.053 versus 1.026, Q3: 1.07 versus 1.059, Q4: 1.097 versus 1.072. There might be reasons why the findings are not as clear in men as in women: less men affected, additional confounders like smoking, occupational exposures, etc. But the point estimates are even higher in men than in women. So from the data I would not argue that men are less susceptible to air pollution.
